# Mixed Corticomedullary Tumor of the Adrenal Gland: A Case Report and Literature Review

**DOI:** 10.3390/medicina59091539

**Published:** 2023-08-25

**Authors:** Jhansi Maradana, Dinesh Edem, Lakshmi Menon, Sonu Abraham, Pruthvi Velamala, Nitin Trivedi

**Affiliations:** 1Endocrinology and Diabetes Consultants, Wentworth Douglass Hospital, 10 Members Way #400, Dover, NH 03820, USA; 2Division of Endocrinology and Diabetes, UAMS, Little Rock, AR 72205, USA; dedem@uams.edu (D.E.); lpmenon@uams.edu (L.M.); 3Lahey Hospital and Medical Center, 41 Burlington Mall Road, Burlington, MA 01805, USA; drsonuabraham@gmail.com; 4Frisbie Memorial Hospital, 11 White Hall Road, Rochester, NH 03867, USA; pruthvi.velamala@gmail.com; 5Saint Vincent Hospital, 123 Summer Street, Worcester, MA 01608, USA; nitin.trivedi@stvincenthospital.com

**Keywords:** mixedcorticomedullary tumor, adrenal tumor, pheochromocytoma, Cushing’s syndrome

## Abstract

Adrenal mixed corticomedullary tumors (MCMTs) are composed of an admixture of cortical and medullary cells. Owing to the presence of two distinct components of different embryonic lineage, these tumors are extremely rare. Less than 30 tumors of this type have been reported to date. MCMTs have varied presentation including hypertension, Cushing syndrome or even as adrenal incidentalomas. Also noted is a slightly higher female preponderance. We report a case of a 26-year-old female who was evaluated for uncontrolled hypertension. A renal ultrasound followed by MRI abdomen revealed a 9.3 × 8.1 × 7.0 cm partially cystic, partially solid enhancing mass in the region of/replacing the left adrenal gland. Hormonal work-up was significant for elevated catecholamines concerning pheochromocytoma. She underwent laparoscopic left adrenalectomy, with adequate pre-operative adrenergic blockade. Histology and immunochemical testing were consistent with a mixed corticomedullary tumor. She was monitored annually for recurrence of the tumor. We also performed a comprehensive review of literature of the cases published so far to the best of our knowledge.

## 1. Introduction

The adrenal cortex is derived from the dorsal mesentery, which is mesodermal in origin, and the medulla from cells that have migrated from the neural crest, which are ectodermal in origin [1]. Most adrenal neoplasms that arise from the adrenal glands are of a single embryonic lineage. Composite or mixed adrenal tumors are very rare, with mixed corticomedullary tumors being the rarest. Mixed corticomedullary tumors comprise a closely intermingled population of adrenal cortical cells and pheochromocytes, each showing distinct histological features of their respective neoplasms. Mixed corticomedullary tumors were first described by Mathison et al. in 1969 in a patient who presented with Cushing’s syndrome and hypertensive crisis during surgery [2]. There was a pre-operative elevation of cortisol and urinary catecholamines with a histopathological exam disclosing mixed adrenal cortical and medullary cells.

We describe a distinctive case of a young woman who presented with uncontrolled hypertension and had markedly elevated urinary and plasma metanephrines. Imaging revealed a large left adrenal mass, which, following resection, revealed a mixed corticomedullary tumor on histopathological analysis.

## 2. Case

A 27-year-old asymptomatic woman was incidentally found to have a blood pressure of 163/114 mm Hg during a regular office visit. Repeat measurements confirmed a persistently high blood pressure at 183/129 mm Hg, which triggered treatment initially with lisinopril followed by the addition of hydrochlorothiazide by her primary care physician. Her past medical history was significant for an uneventful pregnancy four years prior to presentation. Her family history was negative for endocrine tumors or syndromes. On examination, vitals including temperature, heart rate, and respiratory rate were normal except for the elevated blood pressure; physical examination revealed no abdominal masses. Laboratory data showed leukocytosis (16,000/µL) and a normal basic metabolic panel. A renal artery ultrasound, ordered as an initial work up for secondary hypertension, showed a mass adjacent to the spleen and left kidney with an echogenic, homogeneous texture and a hypoechoic, cystic interior. The entire mass measured 9.7 cm × 7.5 cm × 7.5 cm, with the cystic portion measuring 6.2 cm × 5.8 cm × 5.5 cm. Within 2 weeks of initial presentation, the patient developed episodic dizziness, flushing, palpitations and left flank pain. Her blood pressure continued to be uncontrolled and labetalol was added. An MRI of the abdomen and pelvis revealed a 9.3 × 8.1 × 7.0 cm partially cystic, partially solid enhancing mass with T2 hyperintensity in the region of/replacing the left adrenal gland (Figure 1). Endocrinological work-up revealed strikingly elevated 24 h urine metanephrine, 631 mcg (25–222 mcg), and normetanephrine, 27,047 mcg (40–412 mcg), which was supported by high plasma metanephrine, 105 pg/mL (0–62 pg/mL), and normetanephrine, 11,987 pg/mL (0–145 pg/mL), levels. Regretfully, adrenocortical hyperfunctioning was not tested, as she had no clinical manifestations of Cushing’s syndrome or hyperandrogenism. The aldosterone to renin ratio was also not measured. Her serum calcitonin levels, thyroid hormone levels and thyroid ultrasound were normal.

Taking her clinical presentation and laboratory findings into consideration, a provisional diagnosis of pheochromocytoma was made, and she was initially treated with phenoxybenzamine for adequate alpha adrenergic blockade and metoprolol was later added two days prior to surgery. A left adrenalectomy was performed laparoscopically. Histopathological examination demonstrated predominantly large epithelioid or polygonal cells with amphophilic cytoplasm and mildly atypical vesicular nuclei, focally arranged in nests, suggestive of pheochromocytes. Admixed with this were nests and cords of more uniformly rounded epithelial cells with granular eosinophilic cytoplasm and smaller, more uniform vesicular nuclei indicative of cortical adenoma cells (Figure 2). While the bulk of the lesion was diffusely stained positive for chromogranin (negative for synaptophysin) correlating with medulla [Figure 3A], the cords and strands of more eosinophilic epithelioid cells were stained positive for inhibin, MART-1 (Figure 3B) and SF-1, concurring with the cortical component of the adenoma. These findings were consistent with the diagnosis of an adrenal mixed corticomedullary tumor. The postoperative period was uneventful. A month later the patient’s blood pressure normalized without any anti-hypertensive medication. Her plasma metanephrine (<10 pg/mL), normetanephrine (39 pg/mL) and chromogranin A (<1 nmol/L) also normalized. Genetic testing was negative for VHL, RET oncogene and SDHB gene mutations. Other genes were not tested because of lack of insurance approval. Five months post-surgery, she had a recurrence of symptoms with headaches, visual changes and occasional palpitations for a week. Repeat plasma normetanephrine was slightly elevated (274 pg/mL) but plasma metanephrine (25 pg/mL) and chromogranin A (2 nmol/L) were normal. CT of the chest, abdomen and pelvis, however, showed no significant residual tumor. A PET scan and MIBG scan was considered, keeping the possibility of an additional extra-adrenal tumor in mind but could not be performed because of lack of insurance approval. Her symptoms gradually resolved. As the patient was not on any medications/herbs/supplements that can lead to false-positive results, we attributed the mild elevation of normetanephrine to a laboratory error. Continued surveillance revealed normal metanephrine and chromogranin levels. The patient conceived a year later and had an uneventful pregnancy and delivery. She remains asymptomatic and has normal serum metanephrine levels 2½ years after her initial surgery.

## 3. Discussion

As the two components of the adrenal gland have different embryological origins, it is not surprising that mixed corticomedullary tumors are rare. A total of twenty eight patients with adrenal mixed corticomedullary tumors have been described [2,3,4,5,6,7,8,9,10,11,12,13,14,15,16,17,18,19,20,21,22,23,24]. Of these, only three had malignant mixed corticomedullary tumors [16,23,24] and one [16] among those succumbed rapidly to their disease. With that exception, the prognosis of mixed corticomedullary tumors appears to be good [5,16].

The pathogenesis of mixed corticomedullary tumors is poorly understood. Several hypotheses have been proposed. One of these known as the ‘collision theory,’ proposed a separate embryological origin of the two components of the tumor, with the cortical cells arising from the mesoderm and the pheochromocytes from the neural crest and then fusing to form the tumor [5,16]. Another theory suggested that a mutation in the intra-adrenal portal system leads to proliferation of adrenocortical cells with excessive secretion of cortisol, which eventually stimulates the hyperplasia of pheochromocytes [9]. Additionally, it has been proposed that genetic mutations in stem cells can lead to these tumors [18]. Excess secretion of cortisol from cortical tumors has been thought to induce a rate limiting enzyme in catecholamine synthesis which in turn mediates medullary hyperplasia [18]. Finally, the disruption of normal cortico–chromaffin cell interactions by unknown mechanisms could result in a trophic stimulation of both cell lineages [13]. It is interesting to note that the genes involved in tumor formation of the adrenal medullary cells have a minimal or no role in the formation of adrenal cortical tumors [13]. Genetic mutations are common in pheochromocytoma paraganglioma syndromes [25] and more so in young subjects with a positive family history, large extra-adrenal tumors and those with multiple tumors. Pheochromocytoma paraganglioma syndromes are categorized into three clusters based on genetic mutations and pathogenic pathways. Cluster 1 includes mutations in the overexpression of vascular endothelial growth factor gene (VEGF). Cluster 2 includes activating mutations of the WNT signaling pathway. Cluster 3 includes abnormal mutations of the tyrosine kinase pathway. Similarly, there are three biochemical phenotypes—Dopaminergic, noradrenergic, and adrenergic phenotypes—which are defined through significant productions of, respectively, 3-methoxytyramine, normetanephrine, and metanephrine relative to the combined production of all three metabolites. Our patient had a predominantly noradrenergic secreting adrenal tumor which typically occurs with genetic mutations of Von Hippel-Lindau (VHL), succinate dehydrogenase iron sulfur subunit B (SDHB), succinate dehydrogenase complex flavoprotein subunit A (SDHA), succinate dehydrogenase complex subunit C integral membrane protein 15 kDa (SDHC), succinate dehydrogenase complex subunit D integral membrane protein (SDHD) and Fumarate hydratase (FH) genes. Genetic mutations involved in adrenergic secreting adrenal tumors include rearranged during transfection (RET), transmembrane protein 127 (TMEM 127) and myc-associated factor X (MAX) genes. Genetic mutations involved in dopaminergic secreting adrenal tumors include succinate dehydrogenase complex subunit D integral membrane protein (SDHD), succinate dehydrogenase iron sulfur subunit B (SDHB) and succinate dehydrogenase complex subunit C integral membrane protein 15 kDa (SDHC). The biochemical distinction could be due to promoter hypermethylation of the Phenylethanolamine N-methyl Transferase (PNMT) gene which converts norepinephrine to epinephrine. Due to lack of insurance approval, complete genetic testing could not be performed in this patient. Noradrenergic tumors (those predominantly releasing norepinephrine) usually are described as having more continuous symptoms as the secretory pathways are less mature compared to those releasing epinephrine. Thus, noradrenergic tumors often continuously release norepinephrine while adrenergic (epinephrine predominant) tumors more often release epinephrine in paroxysms. Clinical presentations of mixed corticomedullary tumors have varied from subclinical to florid Cushing’s syndrome [3,9,13,17,19,20,22], hypertension [2,3,9,10,13,19,20,21,23,24,26] and diabetes [2,3,13,21] to psychic irritability [3,19] and weight loss [2,3,23]. There is clearly a female preponderance, as is illustrated in Table 1. A total of 74% of the patients presented with hypertension while 39% had Cushing’s syndrome. Our patient developed typical paroxysmal symptoms and uncontrolled hypertension, suggestive of pheochromocytoma. Preoperative treatment to control hypertension is extremely essential due to the high risk of intra-operative hypertensive crisis and malignant arrhythmias. Multiple methods were implemented to achieve pre-operative control of blood pressure with the most widely practiced method being initial alpha-blockade at least 10 to 14 days before surgery and beta blockers added on at least 3 to 4 days after adequate alpha blockade. Some centers use calcium channel blockers as an add on to alpha blockade, while very few centers use metyrosine which inhibits catecholamine synthesis by blocking the enzyme tyrosine hydroxylase. Post-operative biochemical surveillance using the same laboratory tests that were obtained pre-operatively is recommended to confirm a surgical cure of the catecholamine-producing tumor. The risk of recurrent disease following resection of a benign pheochromocytoma or paraganglioma is around 15% [27]. Recurrent disease has not been reported among patients who present with a benign mixed corticomedullary tumor. We have included a review of the literature published so far to the best of our knowledge and availability in Table 1. Given the varied presentation of mixed corticomedullary tumors, a complete work up of adrenal incidentaloma with measurement of both cortical and medullary hormones is recommended to prevent missing a diagnosis of subclinical Cushing’s syndrome, pheochromocytoma or primary aldosteronism.

## 4. Conclusions

This case illustrates a rare case of mixed corticomedullary tumor of the adrenal gland that are composed of an admixture of cortical and medullary cells.Owing to the presence of two distinct components of different embryonic lineage, these tumors are extremely rare. Less than 30 tumors of this type have been reported to date.Patients may present with Cushing’s syndrome, hypertension or non-specific symptoms like weight loss.A high degree of suspicion is required to avoid a hypertensive crisis during surgery. For this reason, clinicians should be aware of this entity.

## Figures and Tables

**Figure 1 medicina-59-01539-f001:**
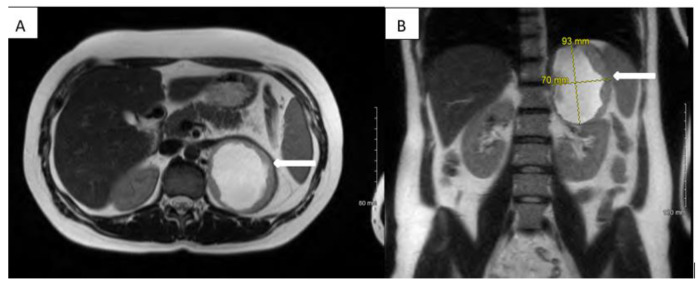
MRI. Axial (**A**) and coronal (**B**) T2 MRI image showing a large hyperintense mass in the region of left adrenal gland.

**Figure 2 medicina-59-01539-f002:**
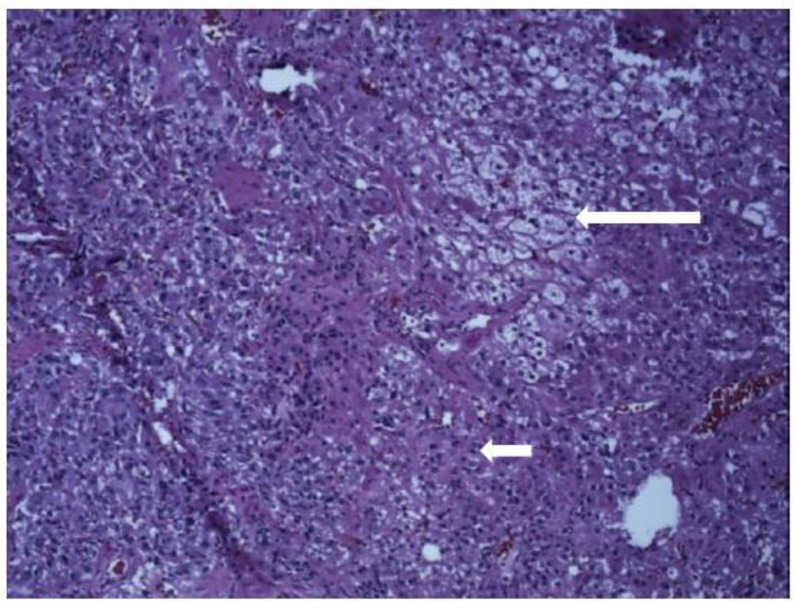
Histology.

**Figure 3 medicina-59-01539-f003:**
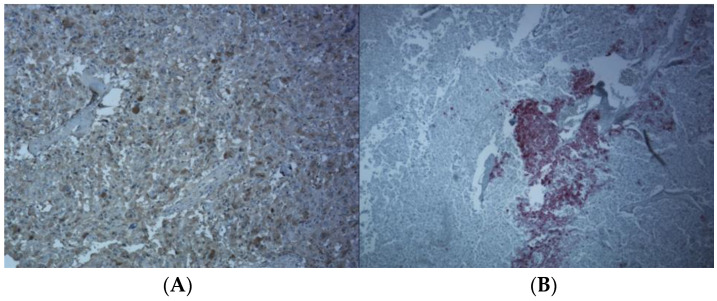
Chromogranin staining.

**Table 1 medicina-59-01539-t001:** Review. This table includes a comprehensive review of literature published on various presentations of mixed corticomedullary tumors.

**Year**	**Age**	**Gender**	**Presenting Symptoms**	**Tumor Size (cm)**	**Cystic Component**	**Hormones**
1969 [2]	39	F	Intraoperative hypertensive crisis, Cushing’s syndrome	4	N/R	Cortisol, catecholamines (both nor-met and met)
1993 [3]	61	F	Hypertension, hyperglycemia	3.5	N/A	Cortisol, catecholamines (nor-met predominant)
1995 [28]	32	M	Cushing’s syndrome	4.5	N/A	Cortisol
1996 [4]	56	F	Cushing’s syndrome, paroxysmal hypertension	8	No	Cortisol, aldosterone, catecholamines (epinephrine predominant)
1996 [4]	32	F	Cushing’s syndrome, Paroxysmal hypertension	9	No	Cortisol, aldosterone, catecholamines (epinephrine predominant)
2001 [5]	34	F	Hypertension, hair loss and amenorrhea	4.5	No	Cortisol
2001 [5]	52	F	Flank pain	2.5	No	None
2002 [6]	N/A	N/A	Hypertension	N/A	N/A	Catecholamines
2002 [6]	N/A	N/A	hypertension	N/A	N/A	Aldosterone
2002 [6]	N/A	N/A	hypertension	N/A	N/A	Aldosterone
2002 [6]	N/A	N/A	hypertension	N/A	N/A	Aldosterone
2003 [7]	55	F	Mild Cushing’s syndrome and hyperglycemia	2.5	No	Cortisol
2007 [8]	41	F	Cushing’s syndrome, intra-operative hypertension	4	No	Cortisol (catecholamines not tested)
2008 [9]	25	F	Gestational diabetes, paroxysmal hypertension, Cushing’s syndrome	3.2	No	Cortisol, catecholamines (Nor-met predmoninant)
2009 [10]	54	F	Mild hypertension, diabetes	4.9	No	Cortisol, Catecholamines (Nor-met predominant)
2009 [11]	66	F	Subclinical Cushing’s syndrome	4.2	Yes	Cortisol
2010 [12]	48	F	Abdominal distension, pedal edema, breathlessness	8	No	Catecholamines (Urinary VMA)
2011 [13]	64	F	hypertension	3.6	No	Catecholamines (metanephrines)
2012 [14]	78	F	Hypertensive urgency	10	No	Catecholamines (dopamine)
2013 [15]	53	M	Incidental finding of left adrenal mass	5.5	No	Cortisol and catecholamines (NSE and chromogranin A)
2013 [16]	63	M	Right upper quadrant palpable mass	8	No	Cortisol
2016 [17]	48	M	Cushing’s syndrome, hypertension	3.9	No	Cortisol, catecholamines (metanephrine)
2017 [18]	58	M	Fluctuant blood glucose levels, paroxysmal hypertension	3	No	Cortisol, Catecholamines (Urinary VMA)
2020 [19]	31	F	Gestational hypertension, psychiatric disturbances persistent to postpartum	3.8	No	Cortisol, Catecholamines (Normetanephrine predominant)
2022 [20]	52	F	Palpitations, weight gain, anxiety	4.1	No	Cortisol, Catecholamines (both met and nor-met)

## Data Availability

Not applicable.

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
