# Peer review of "Mixed Corticomedullary Tumor of the Adrenal Gland: A Case Report and Literature Review"

_medicina, 2023, doi:10.3390/medicina59091539_

Round 1

Reviewer 1 Report

I commend Dr. Maradana and colleagues for this case report and review of the literature. They describe a case of a mixed corticomedullary tumor and I must admit I greatly enjoyed reading this report. I have a few minor comments to enhance this manuscript further for the authors to collectively consider. 

1. Consider a relevant discussion of genetic clusters and biochemical phenotypes (noradrenergic versus adrenergic). Here the authors aptly point out that SDHx mutations, FH, and VHL lead to tumors that predominantly release norepinephrine; however, considering clusters of mutations is more appropriate as is a discussion of biochemical phenotypes.

2. Pursuant to comment 2, noradrenergic tumors (those predominantly releasing norepinephrine) usually are described as having more continuous symptoms as the secretory pathways are less mature compared to those releasing epinephrine. Thus, noradrenergic tumors often continuously release norepinephrine while adrenergic (epinephrine predominant) tumors more often release epinephrine in paroxysms. This is especially true in papers that have investigated the difference between VHL and MEN2A patients. 

3. I agree with the authors that it is unfortunate that the patient did not have additional hormonal evaluation. Nonetheless, this is understandable given insurance issues and the pragmatic realities of clinical practice. I mention this more as "food for thought" rather than implying that it must be discussed. A patient with an admixed tumor that also produced cortisol in addition to catecholamines would theoretically be challenging as cortisol (in fact all steroids) augment PNMT activity and thus increase the metabolic conversion of NE to EPI. Again this is more an intellectual point for the authors to consider rather than a specific tangible need to change anything in the discussion. This patient had paroxysmal symptoms which I think is fine as the patient knows best. Nevertheless, a point/concept to consider here. 

4. In reference to comment 2 consider calculating the biochemical phenotype of the patient based on their metanephrine values as follows (URL - metanephrine)/ (URL-metanephrine) + (URL - normetanephrine) further details can be found in the following manuscript for which this reviewer has no vested interest and is not an author 10.1210/endrev/bnad011  

Reviewer 2 Report

The article titled ‘Mixed Corticomedullary Tumor of the Adrenal Gland: A Case Report and Literature Review’ reports a rare case of young woman who presented with uncontrolled hypertension and had markedly elevated urinary and plasma metanephrines, which is associate to the generation of mixed corticomedullary tumor on histopathological analysis that potentially induced by adrenal. The authors’ finding in their clinical case is rare since adrenal mixed corticomedullary tumors (MCMTs) are composed of an admixture of cortical and medullary cells and the presence of two distinct components of different embryonic lineage make these tumors extremely rare with less than 30 cases had been reported to date.

Overall, I think this article is well organized and written, and the summary is reasonable to me. This case report provides an additional real case scenario to help diagnostic and disease treatment to the patients in the future. I suggest accepting this article.

Need some improvement

Author Response

Thank you for taking your valuable time to review our manuscript.  Please review our revised manuscript.